# Evidence of Virulent Multi-Drug Resistant and Biofilm-Forming *Listeria* Species Isolated from Various Sources in South Africa

**DOI:** 10.3390/pathogens11080843

**Published:** 2022-07-27

**Authors:** Christ-Donald Kaptchouang Tchatchouang, Justine Fri, Peter Kotsoana Montso, Giulia Amagliani, Giuditta Fiorella Schiavano, Madira Coutlyne Manganyi, Giulia Baldelli, Giorgio Brandi, Collins Njie Ateba

**Affiliations:** 1Food Security and Safety Focus Area, Faculty of Natural and Agricultural Sciences, North-West University, Mmabatho 2735, South Africa; christdonaldk@yahoo.com (C.-D.K.T.); frijustine2000@gmail.com (J.F.); montsokp@gmail.com (P.K.M.); 2Department of Biomolecular Sciences, University of Urbino Carlo Bo, 61029 Urbino, PU, Italy; giulia.amagliani@uniurb.it (G.A.); giulia.baldelli@uniurb.it (G.B.); giorgio.brandi@uniurb.it (G.B.); 3Department of Humanities, University of Urbino Carlo Bo, 61029 Urbino, PU, Italy; giuditta.schiavano@uniurb.it; 4Department of Biological and Environmental Sciences, Faculty of Natural Sciences, Walter Sisulu University, Mthatha 5117, South Africa; madiramanganyi@gmail.com

**Keywords:** *L. monocytogenes*, virulence, multi-drug resistance, biofilm formation, food safety

## Abstract

Listeriosis is a foodborne disease caused by *Listeria monocytogenes* species and is known to cause severe complications, particularly in pregnant women, young children, the elderly, and immunocompromised individuals. The aim of this study was to investigate the presence of *Listeria* species in food and water using both biochemical and species-specific PCR analysis. *L. monocytogenes* isolates were further screened for the presence of various antibiotic resistance, virulence, and biofilm-forming determinants profiles using phenotypic and genotypic assays. A total of 207 samples (composed of meat, milk, vegetables, and water) were collected and analyzed for presence of *L. monocytogenes* using species specific PCR analysis. Out of 267 presumptive isolates, 53 (19.85%) were confirmed as the *Listeria* species, and these comprised 26 *L. monocytogenes*, 3 *L. innocua*, 2 *L. welshimeri*, and 1 *L. thailandensis*. The remaining 21 *Listeria* species were classified as uncultured *Listeria*, based on 16SrRNA sequence analysis results. A large proportion (76% to 100%) of the *L. monocytogenes* were resistant to erythromycin (76%), clindamycin (100%), gentamicin (100%), tetracycline (100%), novobiocin (100%), oxacillin (100%), nalidixic acid (100%), and kanamycin (100%). The isolates revealed various multi-drug resistant (MDR) phenotypes, with E-DA-GM-T-NO-OX-NA-K being the most predominant MDR phenotypes observed in the *L. monocytogenes* isolates. The virulence genes *prfA*, *hlyA*, *actA*, and *plcB* were detected in 100%, 68%, 56%, and 20% of the isolates, respectively. In addition, *L. monocytogenes* isolates were capable of forming strong biofilm at 4 °C (%) after 24 to 72 h incubation periods, moderate for 8% isolates at 48 h and 20% at 72 h (*p < 0.05*). Moreover, at 25 °C and 37 °C, small proportions of the isolates displayed moderate (8–20%) biofilm formation after 48 and 72 h incubation periods. Biofilm formation genes *flaA* and *luxS* were detected in 72% and 56% of the isolates, respectively. These findings suggest that proper hygiene measures must be enforced along the food chain to ensure food safety.

## 1. Introduction

*Listeria* species are Gram-positive bacteria that belong to *Listeriacaeae* family. They are widely distributed in nature and capable of causing listeriosis in humans and animals [1]. In terms of pathogenicity, *L. monocytogenes* and *L. ivanovii* are the most pathogenic species, and they are responsible for several outbreaks in humans [2]. Their ubiquitous nature allows them to strive in various environments, particularly water, soil, plants, animals, silage, sewage, and food processing environments [3]. In the food industry, *L. monocytogenes* is regarded as a significant pathogen of clinical importance and responsible for food contamination worldwide [4]. Despite this, listeriosis is underreported, especially in developing countries, and this is a serious concern, given that it accounts for nearly 20–30% mortality in pregnant women, newborn babies, immunocompromised patient, and the elderly [5]. In humans, symptoms include vomiting, flu-like illness, diarrhoea, fever, and abdominal pain [6]. Furthermore, pregnant women are reporting symptoms such as headaches, chills, dizziness, and gastroenteritis, which can lead to stillbirth, early delivery, abortion, or septicemia in the child [7]. In addition, listeriosis may lead to other fatal conditions, such as encephalitis and rhombencephalitis, and these may result to meningoencephalitis, conjunctivitis, pneumonia, and meningitis, especially in infants [8,9,10].

When compared to other common bacterial species, *L. monocytogenes* does not cause foodborne diseases as frequently. Because of the severity of its effects on vulnerable persons, as well as the potential to produce disproportionally high levels of morbidity and death in patients, it is a pathogen of great public health concern. [11]. Moreover, an increasing trend in the occurrence of human invasive listeriosis, even in developed countries, which have advanced public health facilities, is a cause for concern [12]. Several studies have reported the presence of *Listeria* spp. in food and food products, including vegetables, pre-cooked or frozen meat, poultry, pork, fresh produce, dairy products, seafood, and ready-to-eat (RTE) foods [13,14,15,16]. Owing to current lifestyles, individuals rely heavily on ready-to-eat (RTE) and ready-to-cook (RTC) food products that have undergone minimal processing, in order to meet their nutritional requirements [17,18]. These bacteria have the potential to increase the risks of food contamination by bacteria pathogens like *Listeria* species, with serious consequences for public health.

Recently, the United States Food and Drug Administration (FDA) published an announcement for several companies reporting a recall of various food, such as enoki mushrooms (Enoki mushroom was imported by Shanghai Finc Food Co., Ltd. Supplier fron Shanghai, China), cream sauce products, salads, fish and pickled fish products, seafood mushroom, jalapeno cream cheese, spicy queso dip, and queso, due to the potential risk of *L. monocytogenes* contamination [19,20,21]. Several countries, such as the United States of America [22], the European Union (Austria, Denmark, Finland, Sweden, the United Kingdom, Australia, and Spain), Japan, and China have experienced massive listeriosis outbreaks [15]. In 2018, South Africa reported the world’s most severe listeriosis outbreak associated with contamination of the processed meat product (polony) and characterized by a gradual increase in cases of the disease from January 2017 to July 2018. Out of the 1060 laboratory-confirmed cases of listeriosis reported by the National Institute of Communicable Diseases (NICD), 216 deaths were recorded [15]. Massive fatalities, with a projected economic cost to the tune of USD 260 million for lethality and USD 10.4 million for hospitalization, were recorded in South Africa [23]. In addition, the company that was implicated as a source of contamination with *L. monocytogenes* suffered an estimated loss of USD 15 million, due to a recall and temporary ban to the production and selling of certain meat products (polony) [23].

The prudent use of antimicrobial agents is critical in the effective management of the infections caused by *L. monocytogenes,* especially for an organism that has historically been susceptible to a number of antimicrobial agents [11]. Although the *Listeria* species are susceptible to quinolones, monobactams, fosfomycin, and broad-spectrum cephalosporins [2], and beta-lactam antibiotics (penicillin, ampicillin), with or without gentamicin, are generally recommended for the treatment of listeriosis. Alternative treatments include vancomycin and trimethoprim/sulfamethoxazole in patients who are allergic to penicillin [24]. However, recent studies have reported the occurrence of antimicrobial resistance (AMR) [25,26] and biofilm-forming *L. monocytogenes* [27,28], including those circulating in food-producing animals [29], food processing environments [30,31], the food chain [11], and healthcare facilities [32]. It is also worth noting that the current life-span of antimicrobial agents has reduced over time [33]. Furthermore, it is well-documented that biofilms contribute to an increase in antibiotic resistance, since biofilm-forming bacteria have been reported to be more resistant to antibiotics, when compared to their planktonic counterparts [34]. The biofilm structure (extracellular polymeric substance) enhances the ability of microbial cells to withstand and survive chemical compounds, including those used in food industry [35,36]. For this reason, understanding the pathogenicity of *L. monocytogenes*, through the determination of the presence of virulence genes that enable entry, proliferation, and spread in host cells, is critical [37,38,39]. Amongst them, the *inlAB* internalisation locus, *Listeria* pathogenicity island-1 (LIPI-1), *hpt* intracellular growth locus, *InlB, prfA, plcA, hly, mpl, actA*, and *plcB*, genes have been associated with virulence in *L. monocytogenes* [40,41,42,43,44,45]. Some studies have also reported a reciprocal association between antibiotic resistance and increased bacterial virulence and transmission [46,47]. Therefore, the aim of the current study was to investigate the occurrence, AMR, virulence, and biofilm-forming potentials of *Listeria* sp. isolated from various sources in the North West province of South Africa.

## 2. Results

### 2.1. Detection of Listeria spp.

Out of the 207 samples screened, a total of 267 isolates were presumptively positive for *Listeria* spp., based on the morphological and biochemical characteristics (Gram-positive rods, catalase-positive, beta hemolysis, H_2_S-negative, gas-negative, oxidase-negative, methyl red-positive, and esculin hydrolysis) of the isolates. Of these, 53 (19.85%) were confirmed positive for the genus *Listeria* spp. using PCR analysis. These constituted 14 (5.24%) from water samples, 36 (13.48%) from food samples and 3 (1.12%) from fecal samples. Based on species specific PCR assays, 25 (47.17%) isolates were confirmed as *L. monocytogenes*, 14 (26.41%) from water samples, 3 (1.12%) from vegetables, and 8 (15.09%) from meat/meat products) (Table 1). However, the 16S rRNA sequence BLAST search results revealed that of the 53 *Listeria* spp. isolates sequenced, 26 were identified as *L. monocytogenes*, while the others were 3 *L. innocua*, 2 *L. welshimeri*, 1 *L. thailandensis*, and 21 uncultured *Listeria* spp.

### 2.2. Antibiotic Resistance

#### 2.2.1. Antibiotic Resistant Profiles of *L. monocytogenes* Strains

A total of 25 *L. monocytogenes* isolate were subjected to antimicrobial susceptibility test against 12 antibiotics. The results revealed that 62.5% to 100% of *L. monocytogenes* isolates were resistant to most of the antibiotics, including ampicillin, trimethoprim and sulfamethoxazole, erythromycin, and norfloxacin. Twenty-four isolates were resistant to kanamycin. Most of the isolates were found to be resistant to a total of seven antibiotics (kanamycin, novobiocin, tetracycline, oxacillin, clindamycin, nalidixic acid, and gentamicin) tested. The isolates from water samples were more resistant than those isolated from food samples. Table 2 and Table 3 showed the proportion of the AMR phenotypes of the isolates. Thirteen MAR phenotype patterns were observed in *L. monocytogenes* isolates, with E-DA-GM-T-NOV-OX-NA-K being the most frequently detected phenotype. The MAR indices ranged from 0.6 to 0.9, with a mean of 0.74 (Table 3).

#### 2.2.2. Antibiotic Resistance Genes

A total of 25 *L. monocytogenes* isolate were screened for the presence of AMR determinants. Out of the twenty AMR genes tested, 10 were detected in 23 isolates. The highest detections were *ant (3″)-la* (92%, 23/25), *dfr1S* (84%, 21/25), *sul1* (80%, 20/25), and *ermC* (80%, 20/25), while the lowest were *tetA* and *tetB* at 8% (2/25) (Figure 1).

### 2.3. Proportion of Virulence Genes Detection in L. monocytogenes 

Out of eight virulence genes, five (*prfA*, *actA, hlyA, inlC, inlJ*) were detected in 25 *L. monocytes* by PCR. The *prfA* was the most frequently (100%) detected gene in all *L. monocytogenes* isolates when compared to the *hlyA* (68%, *n* = 17) and *actA* (56%, *n* = 14) genes. Only five isolates harbored the *plcA* gene. Out of 25 *L. monocytogenes* isolates, 3 from food samples carried 8 virulence genes. In general, 5 virulence genes (*actA, hlyA, inlC, inlJ, prfA*) were most often detected in this study (Appendix A).

### 2.4. Biofilm-Forming Potential of L. monocytogenes Isolates

#### 2.4.1. Phenotypic Assessment of Biofilm Formation 

The microtiter plate assay was employed to assess the ability of *L. monocytogenes* isolates to form biofilm at different temperatures (4, 25, and 37 °C) over 24, 48, and 72 h periods. Based on biofilm formation patterns, the isolates were grouped either as non-adherent, weakly, moderately, or strongly adherent strains (Table 4). The results showed that all the isolates (*n* = 25) exhibited strong biofilm attachments, following 24, 48, and 72 h incubation periods at 4 °C. At 25 °C, 2 (8%) *L. monocytogenes* isolates produced moderate biofilm, while 23 (92%) formed a strong biofilm attachment, following 24 and 72 h incubation periods. Strong biofilm formation was demonstrated by 20 (80%) isolates at 37 °C after a 48 h incubation period. At 37 °C, the production of biofilm was strong after a 24 h incubation period, and the biofilm production of some isolates decreased after 72 h (*p* < 0.05), as shown in (Appendix B) Figure A1, Figure A2 and Figure A3.

#### 2.4.2. Biofilm Formation Genes

PCR was utilized to determine the presence of genes associated with biofilm formation. Out of 25 *L. monocytogenes* isolates, 14 were positives for *luxS* and *flaA* (56% and 72%, respectively). A total of 9 (36%) isolates from food samples carried *luxS* gene followed by 5 (20%) from water and food samples harboring *flaA* gene. A total of 9 isolates (5 from food samples and 4 from water samples) were positive for both *flaA* and *luxS* genes (Appendix A). 

## 3. Discussion

*L. monocytogenes* is capable of causing a sporadic disease, termed listeriosis, in humans and animals that results in very high hospitalization and fatalities [48]. Listeriosis is amongst the most problematic food-borne diseases in several countries, and it is associated with contamination at all stages in the food production process [49,50,51,52]. This pathogen is a psychrophile and can grow at refrigerated temperatures [53,54,55]. Due to this feature, it is very difficult to inactivate and curb the proliferation of *L. monocytogenes* cells in RTE food. This, therefore, indicates the need for continuous monitoring of this pathogen, especially in the food industry [56]. The main objective of this study was to assess the presence of *Listeria* sp. from various sources that are linked to the food chain, especially after South Africa recorded the world’s most severe outbreak of listeriosis in 2017 [57]. The results obtained in this study show that the occurrence of *Listeria* spp. was higher in food products (13.48%) than water (5.24%) and cattle feces (1.12%) samples. *L. monocytogenes* was detected in water and food. Interestingly, *L. monocytogenes* was most often detected in meat and meat products (eight; 15.09%), when compared to vegetables (three; 5.66%). These findings are a cause for concern, given the fact that listeriosis outbreaks have been associated with meat and poultry products, such as hot dogs [58], RTE pork meats [59], and turkey meat products [60]. In addition, *L. monocytogenes* was most frequently isolated in packing houses that processed frozen (41.3%), fresh-cut vegetables (7.9%) [61,62] and vegetables (9.5%) [63], mushrooms (1.6%) [64], and prepackaged salads and canned vegetables (5.4%) [65]. Food products tested in this study, such as cucumber and lettuce, may have been contaminated with *L. monocytogenes* as a result of soil contact. This pathogen may be introduced into food processing plants by human carriers (workers) and animals through faecal shedding [66,67]. The risks that these bacteria (*L. monocytogenes)* provide to public health are highlighted by their ability to survive for years, or even decades, in food processing facilities, as well as their ability to assist in food contamination during processing, handling, and packaging and the ease of foodborne transmission to consumers. Moreover, the detection of *L. monocytogenes*, including other species, such as *L. ivanovii, L. grayi,* and *L. innocua*, in wastewater effluents in a treatment facility in South Africa clearly indicates the potential to contaminate the environment. This may also increase the routes of transmission and dissemination of *L. monocytogenes* to foods [68].

Another objective of this study was to determine the antimicrobial resistant profiles of *L. monocytogenes* isolated from different sources within the food chain. In this study, large proportions of *L. monocytogenes* isolates were resistant to the majority of the antibiotics (62.5% to 100%). These results are consistent with another study, which reported high (90.74%) resistant *L. monocytogenes* isolates against nalidixic acid [69]. MAR (Multi-antibiotic resistance is a bacterium that is resistant to several antibiotics) were most prominent with nalidixic acid/cloxacillin (35.2%), nalidixic acid/colistin (31.5%), and cloxacillin/colistin/nalidixic acid (29.6%) [70]. However, a low (5.6%) AMR pattern was shown in chloramphenicol/nitrofurantin/cotrimoxazole [61]. In this study, the MAR phenotype E-DA-GM-T-NOV-OX-NA-K appeared in three different locations on different samples, which could be due to the use of antibiotics in those area. The continuous misuse of antibiotics led to the development of antibiotic resistance, specifically multi-drug resistant (MDR, which is an isolate that is resistant to at least one antibiotic in three or more drug classes) strains [71,72]. Notably, a high MAR index (Mean = 0.74) was observed in this study, and these findings were higher than that of the previous study [73]. This suggest that the isolates originate from a high-risk environment. Furthermore, the results obtained in this study revealed that *L. monocytogenes* harbored antimicrobial resistant genes *ant(3″)-la* (92%), *dfr1S* (84%), *sul1* (80%), and *ermC* (80%), while the lowest were *tetA* and *tetB* at 8%). Contradictory to our findings, Heidarzadeh et al. reported *tetM* was detected at high rates (70%) of tetracycline-resistant isolates [70]. Approximately, 27.3% isolates carried *dfrD*. The *ermB* gene was detected in 83.3% of the erythromycin-resistant isolates [70]. Although antibiotic resistance in *L. monocytogenes* was rare, over the last two decades, there has been an increasing trend in resistance to one or multiple common antibiotics found in both clinical and environmental sources. In general, resistance genes are handed along by transferable plasmids and conjugative transposons, and *L. monocytogenes* resistant to antibiotics is no exception. Despite the fact that the study was unable to prove a direct transmission link between resistant genes in *L. monocytogenes*, our findings revealed several resistance genes in water and food samples.

A further objective of the study was to determine the presence of virulence genes in *L. monocytogenes.* The detection of *prfA* (100%), *hlyA* (68%), and *actA* (56%) genes indicates the potential pathogenicity of the isolates and their potential to cause foodborne infections. In addition, *plcB* genes were detected in five isolates. In contrast to our findings, Du et al. [74] found *plcB* gene in all isolates, whereas the *inlB*, *actA*, *plcA*, and *iap* genes were found in 71.4–90.5% of the isolates. Several studies have shown that the *prfA* gene in *L. monocytogenes* encodes a protein that induces the transcription of the listeriolysin gene (*lisA*) [75,76,77,78]. This signifies the *prfA* gene in the pathogenesis of listeriosis. Furthermore, the *prfA* gene in *L. monocytogenes* possessed two important crucial functions: response to environmental stress conditions and capacity to infect humans [75].

Biofilm formation, as well as the adaptation to stress, induced by cold temperatures and sub-lethal concentrations of disinfectants, coupled with the potential to inhabit ecological niches in facilities and equipment that are difficult to clean, have been identified as key contributors to the persistence of *L. monocytogenes* in the food chain [79]. Biofilm formation or ‘glycocalyx’ facilitates the adherence of bacteria to surfaces and plays a significant role in the success of pathogenic bacteria; thus, the prevention of adhesion could be an effective way to combat infections and cross-contamination [80]. This study evaluated the biofilm formation potentials of 25 *L. moncytogenes* strains isolated from different sources. *L. monocytogenes* has the potential to multiply within a wide range of temperatures (2–45 °C); due to that, three various temperatures were evaluated, which comprised 37 °C (representing the optimum growth temperature range between 30 and 37 °C), 25 °C (representing the room temperature), and 4 °C (which stand for refrigeration temperature of foods in retail) [15,81,82,83]. Our findings reveal that strong biofilm attachments were formed at 4 °C for 24, 48, and 72 h in all isolates (*n* = 25; 100%), followed by moderate biofilm formation for two (8%) isolates at 48 h and five (20%) *L. monocytogenes* isolates at 72 h. Similar results by Lado et al. [81] showed that *L. monocytogenes* isolates produced a strong biofilm community at 10 and 37 °C. All *L. monocytogenes* isolates utilized in this study were from food sources, as well as food processing equipment. 

In addition, PCR-based biofilm formation genes showed that 18 out of 25 (72%) were amplified for *flaA*, followed by 14 (56%) for *luxS*. A total of nine (36%) isolates from food samples were found to carry the gene *luxS*, followed by five (20%) from water and food samples harboring the *flaA* gene. In a recent study, it was proven that *L. monocytogenes* biofilm was produced at optimum of neutral pH and 37 °C; supplementary data showed temperature as a key player in *L. monocytogenes* biofilm formation. It was concluded that the impacts of various factors on biofilm production are strain-dependent [74]. In Egypt, particularly Sohag city, 295 samples of ice cream, vaginal swabs, stool, and the urine of aborted women were collected and examined for the virulence ability of *L. monocytogenes*. Biofilm investigation was conducted by PCR detection of biofilm genes, such as *luxS* and *flaA*. The *luxS* gene was detected at 81.8%, followed by *flaA* at 72.7% in ice cream. In aborted women, the *luxS* gene amplified 81.8%, followed by *flaA* (63.6%), which contradicted our results [84]. In another study, Hassanien and Shaker assessed the biofilm potential of *L. monocytogenes* and showed that out of 38 isolates amplified, 23 (60.52%) and 1 (2.63%) were positive to *luxS* and *flaA* genes respectively [85].

Over the past two decades, researchers have shown the impact of biofilm in the food processing industries. These biofilms contribute to cross contamination by protecting and harboring harmful or spoilage microorganisms, including *L. monocytogenes* [86]. Antibiotic resistance is higher in the context of biofilm development than in planktonic cells [35,87]. Hence, this poses a problem in the control of *L. monocytogenes* from food sources, water, and food processing surfaces, as well as equipment.

## 4. Materials and Methods

### 4.1. Sample Area and Sample Collection

A total of 207 samples were collected randomly from different sources (water, meat, vegetable, milk, and cattle feces) around the North West province in South Africa, between June 2018 and March 2020. The samples comprised 59 water (42 from boreholes and 17 from rivers) and 47 meat (12 wors (a South African traditional home-made sausage that is commercially available), 10 Russian (a South African sausage-shaped packed meat with a ‘strong meaty flavour and an intense smoky finish), 10 mince, and 15 red meat) samples. In addition, 23 vegetable samples (comprising 6 carrots, 5 cucumber, 4 lettuce, 4 cabbage, and 4 spinach), 34 milk (22 pasteurized and 12 unpasteurized), and 44 cattle feces samples were collected. The water samples were collected from eight sites, including Mafikeng, Potchefstroom, Klerksdorp, Zeerust, Vryburg, Marikana, Lichtenburg, and Groot Marco (Figure 2). For the collection of cattle feces, aseptic procedures were followed, and samples were obtained directly from the rectum of the animals, using sterile arm-length gloves, which were changed for each sample collected. The food samples were collected from the shelves of different shops and displayed for consumption by the consumers. Non-pasteurized milk was collected from dairy farms during the milking of cattle and distributed into sterile 250 mL bottles. All samples were transported on ice to the laboratory and processed within 24 h of collection. Figure 2 shows the different areas from which samples were collected.

### 4.2. Bacterial Strains

The reference strains *L. monocytogenes* (ATCC 19115) and *L. ivanovii* (ATCC 19119) were used as positive controls, while *Escherichia coli* (ATCC 25922) was used as negative control in the study. These American-type culture collection strains were purchased from Davies Diagnostics, Randburg, South Africa.

### 4.3. Isolation of Listeria spp.

To process the fecal, meat, and vegetable samples, 25 g of each sample was homogenized in 225 mL of half-Fraser broth (HFB, Acumedia, Lansing, MI, USA), while, for milk samples, 25 mL of each sample was transferred into HFB. Aliquots of 100 mL of water samples were filtered through a 0.45-μm Whatman filter paper (Whatman^®^ glass microfiber GS filter paper, Separation Scientific SA (Pty) Ltd., Johannesburg, South Africa) using a vacuum pump machine (model N035AN.18, Freiburg, Germany). The membrane filters were inoculated in 225 mL of HFB and incubated aerobically at 30 °C for 24 h. Following incubation, aliquots of 100 µL from each sample were transferred into 10 mL of Fraser broth (Acumedia) and incubated at 30 °C for 24 h. Tubes with black cultures, resulting from the hydrolysis of esculin, were considered to be presumptively positive for *Listeria* species, and 100 µL aliquots were spread-plated on *Listeria* isolation medium (Oxford formulation, Acumedia) [88]. The plates were incubated at 30 °C for 24–48 h. Pure isolates with greenish-grey morphologies and a concave center surrounded by a black halo (due to hydrolysis of esculin) were selected for further identification tests. Two to three colonies were picked from each plate and subjected to Gram staining and biochemical testing, such as the fermentation of carbohydrates (mannitol, rhamnose, dextrose, maltose, and xylose), motility, catalase activity, and β-hemolysis [89]. Samples with positive β-hemolysis and catalase tests fermented most sugars, including rhamnose, dextrose, and maltose, but not mannitol or xylose, were identified as *Listeria* spp. and sent to molecular identification for confirmation.

### 4.4. Molecular Characterizations of Listeria Species Using PCR Analysis

#### 4.4.1. Extraction of Chromosomal DNA

Genomic DNA of each presumptive isolate was extracted using a commercial Genomic DNA^TM^ Tissue Miniprep Kit (Zymo Research, Irvine, CA, USA), following the manufacturer’s instructions; the quantification of the DNA was performed using a NanoDrop^TM^ 1000 spectrophotometer (Thermo Fischer Scientific, Waltham, MA, USA).

#### 4.4.2. Identification of *Listeria* spp., Virulence Genes, Antibiotic Resistance Genes and Biofilm Genes Using PCR (Polymerase Chain Reaction) Technique

Except otherwise mentioned, all PCR reagents used in this study were from Fermentas (Waltham, MA, USA), and all primers were synthesized by Inqaba Biotechnical (Pty) Ltd. (Pretoria, South Africa). DNA was subjected to bacterial 16S rRNA gene amplification, and the amplicons were sent to Inqaba Biotechnical (Pty) Ltd. (Pretoria, South Africa) for the 16S rRNA sequencing identification. The *prs* gene was also used for the detection of *Listeria* spp., and the primers (*jograyi*, *lin0464*, *liv22228*, *lmo1030, iseelin* and *lwe180*1) were used for the detection of *Listeria* specific species (Appendix A). DNA from *L. monocytogenes* ATCC 19115 and *L. ivanovii* ATCC 19119 were used as positive controls, while nuclease-free water was used as a negative control. The C1000 Touch^TM^ DNA thermal cycler (Bio-Rad, Singapore) was used to perform the amplifications. PCR primer sequences, target genes, amplicon sizes, and cycling conditions are shown in Appendix A.

PCR was used to detect the actin (*actA*), p60 (*iap*), phosphatidylinositol phospholipase C (*plcA*, *plcB*), internalin (*inla*, *inlb*, *inlc*, *inlj*), regulatory (*prfA*), and hemolysin (*hlyA*) virulence genes, as previously described [90,91,92]. Amplification conditions of each PCR reaction are listed in Appendix A.

The isolates confirmed to be *Listeria* spp. that were presenting resistance to more than two antibiotics were considered as multi-drug resistant (MDR) and were tested for harboring the *erm(A)*, *erm(B1)*, *erm(B2)*, *erm(C)*, *erm(T)*, *bla_TEM_*, *drf1*, *drf5*, *drf12*, *dfr1S*, *dfr5S*, *dfr7s*, *dfr12s*, *dfr17s, tetA*, *tetB*, *ant(3’’)-la*, *sul1*, *sul2* and *pse1* antibiotics resistance genes. The primer sequences, PCR cycling conditions, and amplicon sizes are indicated in Appendix A.

PCR was utilized to detect the biofilm-forming genes *luxS* and *flaA* of *L. monocytogenes* (Appendix A). Each PCR reaction mix was constituted by 12.5 µL of 2X DreamTaq green master mix, 11 µL of RNase nuclease-free PCR water, 0.5 µM of each primer (forward and reverse), and 1 µL of template DNA, for total of 25 µL standard volume. The cycling conditions of the PCR are stated in Appendix A.

All PCR amplicons were separated by electrophoresis on a 2% (*w/v*) agarose gel comprising 0.001 µg/mL ethidium bromide. A 100 bp DNA molecular weight marker (Fermentas) was incorporated in each gel run. Electrophoresis was conducted at 60 Volt for 120 min and the images captured using a ChemiDoc Imaging System (Bio-Rad, UK).

### 4.5. Antimicrobial Susceptibility Test

The Kirby-Bauer disc diffusion technique was used to determine the antimicrobial susceptibility profiles of the isolates [93]. A total of 12 antibiotics (Mast Diagnostics, UK) were used, including ampicillin (AMP), 10 µg; erythromycin (E), 15 µg; kanamycin (K), 30 µg; norfloxacin (NOR), 10 µg; novobiocin (NB), 30 µg; tetracycline (TET), 30 µg; oxacillin (OX), 5 µg; clindamycin (DM), 2 µg; nalidixic acid (NA), 30 µg; gentamicin (GM), 15 µg; meropenem (MEM), 10 µg; and trimethoprim and sulfamethoxazole (SXT), 1.25 + 23.75 µg. The selection of these antibiotics was based on the treatment regimen for listeriosis or their extensive use as prophylactics in beef and dairy cattle farming in South Africa. Overnight pure cultures were prepared in 15mL TSB (tryptic soy broth) and incubated at 37 °C for 24 h. The bacterial suspension was adjusted by performing serial dilution until reaching 10^6^ cells/mL on the McFarland standards scale of 0.5. After 24 h, the bacterial cell suspension were spread on Mueller Hinton agar (Lasec SA (Pty) Ltd, Johannesburg, South Africa) plates using a sterile swab, the plates were incubated aerobically for 24 h at 37 °C, and the diameter of the zones of inhibition around the discs were measured [94]. The results were interpreted based on the antibiotic growth inhibition zone diameter critical values, as recommended by the Clinical Laboratory Standards Institute (CLSI) guidelines [95], in order to classify the isolates as sensitive, intermediate, or resistant to a particular antibiotic. For quality control, *L. monocytogenes* ATCC 19115 and *L. ivanovii* ATCC 19119, were used as reference.

### 4.6. Biofilm Formation

#### Phenotypic Detection

Biofilm formation by *L. monocytogenes* was determined using the microtiter plate assay described by Bertrand et al. [96] and Mahm [97] with slight modifications. *L. monocytogenes* strains were sub-cultured in tryptic soy broth (TSB) supplemented with 0.6% yeast extract (TSBYE) and incubated at 37 °C overnight. A 10 µL aliquot of each overnight culture at 10^5^ CFU was used to inoculate 96-well plates containing 190µL of brain–heart infusion (BHI) broth (Lasec SA (Pty) Ltd, Johannesburg, South Africa) per well. Each isolate was loaded in triplicates. Two positive controls, *Pseudomonas aeruginosa* (strong biofilm producer) and *L. monocytogenes* ATCC 19115 were included, while a negative control well contained 200 µL of BHI only. For all samples, plates were incubated at three different temperatures (4, 25, and 37 °C) for 24, 48, and 72 h. After incubation, the plates were rinsed three times with PBS (300 µL, pH 7.4) to remove unattached microbial cells, and excess PBS was removed by blotting the plate using a clean paper towel and allowed to dry. The dye of crystal violet (0.1%, 200 µL) was added to each well to stain the biofilms formed by the cells on well bottom and incubated at room temperature for one hour. The wells were rinsed three times with 300 µL of PBS, and aliquots of 95% ethanol (200 µL) were added to each well for stain fixation, with incubation at room temperature for an hour. The microplate reader (Shenzhen Heales Technology Development Co., Ltd., Shenzhen, China) was used to read the optical density at 630 nm. The cut-off value of the negative control (ODc) was determined as three standard deviations (SD) above the mean OD of the negative control. As stipulated by Stepanovi’c et al. (2000), biofilm formation was categorized into four groups, according to the ODs obtained: OD ≤ ODc, non-adherent; ODc *<* OD ≤ 2 x ODc, weak biofilm formation; 2 x ODc *<* OD ≤ 4 x ODc, moderate biofilm formation; and 4 x ODc *<* OD, strong biofilm formation.

## 5. Conclusions

The current study revealed the occurrence of *Listeria* spp. from various sample types. Our finding showed that *L. monocytogenes* carried virulence genes and was resistant to several antibiotics. In addition, further investigation indicated strong biofilm formation by *L. monocytogenes* strains. This clearly suggests that *L. monocytogenes* have all the features that contribute to pathogenicity, while being difficult to control, due to the resistance. Hence, effective alternative measures (such as greater attention to good manufacturing and hygiene practices throughout the food production, distribution, and storage chains, including in the home environment) are needed to control and prevent resistant *L. monocytogenes* and reduce the risk of listeriosis.

## Figures and Tables

**Figure 1 pathogens-11-00843-f001:**
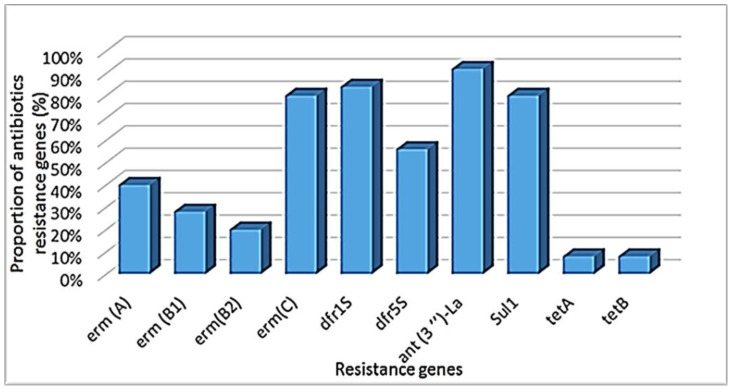
Proportion of antibiotic resistance genes detected in *L. monocytogenes* strains.

**Figure 2 pathogens-11-00843-f002:**
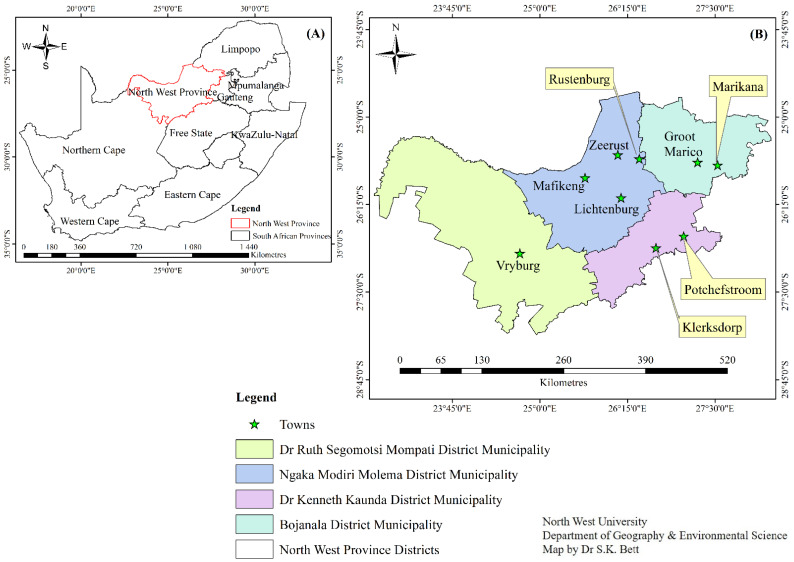
The map where the samples were collected ((**A**) represents the map of the North West province, and (**B**) represents the map of the districts and towns).

**Table 1 pathogens-11-00843-t001:** Proportion of *Listeria* spp. isolated from different sample sources.

Sample Source	Number Analyzed	*Listeria* spp. (%)	*L. monocytogenes* (%)
Vegetables	8	3 (1.12)	3 (5.66)
Meat/meat products	108	33 (12.36)	8 (15.09)
Water	143	14 (5.24)	14 (26.41)
Cattle feces	8	3 (1.12)	0 (0)

**Table 2 pathogens-11-00843-t002:** Proportion of *L. monocytogenes* isolates resistant to different antibiotics used in this study.

Isolate Source		Sampling Site	AP	TS	E	MEM	DA	GM	NOR	T	NOV	OX	NA	K
MEAT/MEAT PRODUCTS	Meat	Lichtenburg *n* = 4	1	0	2	1	4	0	4	4	4	4	4	4
	Mince	Lichtenburg *n* = 2	0	1	2	1	2	2	1	2	2	2	2	2
		Zeerust *n* = 2	1	1	2	1	2	2	0	2	2	2	2	2
	Total	N = 8	2(25)	2(25)	6(75)	3(37)	8(100)	4(50)	5(62.5)	8(100)	8(100)	8(100)	8(100)	8(100)
VEGETABLES	Cucumber	Zeerust *n* = 2	0	1	1	1	2	2	0	2	2	2	2	2
	Lettuces	Zeerust *n* = 1	0	0	1	0	1	1	0	1	1	1	1	1
	Total (%)	N = 3	0(0)	1(33.33)	2(66.67)	1(33.33)	3(100)	3(100)	0(0)	3(100)	3(100)	3(100)	3(100)	3(100)
WATER	Borehole	Lichtenburg *n* = 3	1	1	3	3	3	3	3	3	3	3	3	3
		Vryburg *n* = 3	0	1	2	2	3	3	0	3	3	3	3	3
		Mafikeng *n* = 5	1	4	3	0	5	5	0	5	5	5	5	1
	Dam	Mafikeng *n* = 2	0	0	2	0	2	2	0	2	2	2	2	2
		Zeerust *n* = 1	0	0	1	0	1	1	0	1	1	1	1	1
	Total	N = 14	2(14.30)	6(42.86)	11(78.57)	5(35.71)	14(100)	14(100)	3(21.43)	14(100)	14(100)	14(100)	14(100)	10(71.43)

Ampicillin (AP), erythromycin (E), kanamycin (K), norfloxacin (NOR), novobiocin (NOV), tetracycline (T), oxacillin (OX), clindamycin (DA), nalidixic acid (NA), gentamicin (GM), meropenem (MEM), trimethoprim/sulfamethoxazole (TS). The values in brackets represent the resistance percentage.

**Table 3 pathogens-11-00843-t003:** Multi-antibiotic resistant (MAR) phenotypes and MAR index of *L. monocytogenes* isolates.

Resistance Pattern	MAR Phenotype	Number of Observed	MAR Index
I	E-DA-GM-T-NOV-OX-NA-K	7	0.7
II	MEM-DA-GM-T-NOV-OX-NA-K	2	0.7
III	AP-DA-GM-T-NOV-OX-NA-K	2	0.7
VI	TS-E-MEM-DA-GM-NOR-T-NOV-OX-NA-K	1	0.8
V	AP-TS-E-MEM-DA-GM-T-NOV-OX-NA-K	1	0.9
VI	TS-E-MEM-DA-GM-T-NOV-OX-NA-K	1	0.8
VII	DA-GM-T-NOV-OX-NA-K	1	0.6
VIII	AP-E-MEM-DA-GM-T-NOV-OX-NA-K	1	0.8
IX	E-MEM-DA-GM-T-NOV-OX-NA-K	2	0.75
X	TS-E-DA-GM-T-NOV-OX-NA-K	2	0.75
XI	TS-DA-GM-T-NOV-OX-NA-K	1	0.7
XII	TS-E-DA-GM-T-NOV-OX-NA	1	0.7
XIII	TS-E-DA-T-NOV-OX-NA-K	1	0.7

Ampicillin (AP), erythromycin (E), kanamycin (K), norfloxacin (NOR), novobiocin (NOV), tetracycline (T), oxacillin (OX), clindamycin (DA), nalidixic acid (NA), gentamicin (GM), meropenem (MEM), trimethoprim/sulfamethoxazole (TS), multi-antibiotic resistance (MAR). Mean MAR index = 0.74.

**Table 4 pathogens-11-00843-t004:** Proportion of biofilm formation by *L. monocytogenes* isolates at 4, 25, and 37 °C during 24, 48, and 72 h periods.

Incubation Temperature	Incubation Time (h)	Non Adherent (%)	Weak (%)	Moderate (%)	Strong (%)
4 °C	24	0 (0)	0 (0)	0 (0)	25 (100)
	48	0 (0)	0 (0)	0 (0)	25 (100)
	72	0 (0)	0 (0)	0 (0)	25 (100)
25 °C	24	0 (0)	0 (0)	1 (4)	24 (96)
	48	0 (0)	0 (0)	2 (8)	23 (92)
	72	0 (0)	0 (0)	2 (8)	23 (92)
37 °C	24	0 (0)	0 (0)	1 (4)	24 (96)
	48	0 (0)	0 (0)	5 (20)	20 (80)
	72	0 (0)	0 (0)	1 (4)	24 (96)

## Data Availability

The data presented in this study are available on request from the corresponding author.

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
