# Peer review of "Evidence of Virulent Multi-Drug Resistant and Biofilm-Forming Listeria Species Isolated from Various Sources in South Africa"

_pathogens, 2022, doi:10.3390/pathogens11080843_

Round 1
Author Response
Please find attached the response

Reviewer 2 Report
Dear Authors,
I read the manuscript with interest and I have some comments on it, which I put below:
Title: I missed the reference in the introduction to the title "food chain". The text is more indicative of raw materials and products, although there is also water. The "food chain" also means processing, processing, packaging and storage - there were a few sentences missing on this subject.
Introduction: I ran out of information about the most recent listeriosis incidence in South Africa. Please complete this information in the introduction.
The captions for most of the tables are not correct - they should be worded differently.
Please review the text for small errors (so-called typos), because I found a few unnecessary characters, spaces or repeats of numbers (line 341: "...100 100 μL aliquots...") - such author's correction is necessary.
Spelling of Latin names - italics is required and the generic name is capitalized - there are bugs in many places.
Figure 1. If the title of the drawing is "Proportion of antibiotic resistance genes detected in L. monocytogenes strains." this should not describe the "Number of isolates (%)". Please also remove the word "Resistance genes" from above the graph as it is also below it.
The disc diffusion method, as indicated by the Authors themselves (lines 397-404), consists in measuring the diameter of the zones of inhibition around the discs with antibiotics. The Authors did not include the size of these zones of growth inhibition, but only classified the strains as "sensitive, intermediate or resistant to a particular antibiotic". I believe it is useful to include the results obtained for these growth inhibition diameters (mm), as I know from experience that this is information that is often sought by readers.
References: please check carefully how the abbreviations of the journal names are written: they are written arbitrarily, written in small and capital letters - special rules apply in this regard.
I noticed a lot of items in the literature list that are over 20 years old - they are from the 90s and even earlier. It is worth replacing them with slightly newer articles.
Author Response
Please find attached the response to reviewer
